# Adipose-Derived Mesenchymal Stem Cells-Conditioned Medium Modulates the Expression of Inflammation Induced Bone Morphogenetic Protein-2, -5 and -6 as Well as Compared with Shockwave Therapy on Rat Knee Osteoarthritis

**DOI:** 10.3390/biomedicines9101399

**Published:** 2021-10-05

**Authors:** Jai-Hong Cheng, Chieh-Cheng Hsu, Shan-Ling Hsu, Wen-Yi Chou, Yi-No Wu, Chun-En Aurea Kuo, Tsai-Chin Hsu, Li-Yen Shiu, Shun-Wun Jhan

**Affiliations:** 1Center for Shockwave Medicine and Tissue Engineering, Kaohsiung Chang Gung Memorial Hospital and Chang Gung University College of Medicine, Kaohsiung 833, Taiwan; cjocko@gmail.com (J.-H.C.); t1234@cgmh.org.tw (C.-C.H.); hsishanlin@yahoo.com.tw (S.-L.H.); murraychou@yahoo.com.tw (W.-Y.C.); tsaichin1219@gmail.com (T.-C.H.); 2Medical Research, Kaohsiung Chang Gung Memorial Hospital and Chang Gung University College of Medicine, Kaohsiung 833, Taiwan; 3Department of Leisure and Sports Management, Cheng Shiu University, Kaohsiung 833, Taiwan; 4Department of Orthopedic Surgery, Sports Medicine, Kaohsiung Chang Gung Memorial Hospital and Chang Gung University College of Medicine, Kaohsiung 833, Taiwan; 5School of Nursing, Fooyin University, Kaohsiung 831, Taiwan; 6School of Medicine, Fu Jen Catholic University, New Taipei City 242, Taiwan; yino-wu@yahoo.com.tw; 7Department of Chinese Medicine, Kaohsiung Chang Gung Memorial Hospital and Chang Gung University College of Medicine, Kaohsiung 833, Taiwan; lecherries@gmail.com; 8Cell Therapy Center, E-Da Hospital, Kaohsiung 824, Taiwan

**Keywords:** adipose-derived mesenchymal stem cells, conditioned medium, shockwave, osteoarthritis, cartilage repair

## Abstract

The dose-dependent effects of adipose-derived mesenchymal stem cell-conditioned medium (ADSC-CM) were compared with those of shockwave (SW) therapy in the treatment of early osteoarthritis (OA). Anterior cruciate ligament transaction (ACLT) with medial meniscectomy (MMx) was performed in rats divided into sham, OA, SW, CM1 (intra-articular injection of 100 μL ADSC-CM into knee OA), and CM2 (intra-articular injection of 200 μL ADSC-CM) groups. Cartilage grading, grading of synovium changes, and specific molecular analysis by immunohistochemistry staining were performed. The OARSI and synovitis scores of CM2 and SW group were significantly decreased compared with those of the OA group (*p* < 0.05). The inflammatory markers interleukin 1β, terminal deoxynucleotidyl transferase dUTP nick end labeling and matrix metalloproteinase 13 were significantly reduced in the CM2 group compared to those in the SW and CM1 groups (*p* < 0.001). Cartilage repair markers (type II collagen and SRY-box transcription factor 9, SOX9) expression were significantly higher in the CM2 group than in the other treatment groups (*p* < 0.001; *p* < 0.05). Furthermore, inflammation-induced growth factors such as bone morphogenetic protein 2 (BMP2), BMP5, and BMP6 were significantly reduced in the treatment groups, and the CM2 group showed the best results among the treatments (*p* < 0.05). In conclusion, ADSC-CM and SW ameliorated the expression of inflammatory cytokines and inflammation-induced BMPs to protect the articular cartilage of the OA joint.

## 1. Introduction

Osteoarthritis (OA) is a degenerative joint disease that results in damage to the cartilage and subchondral bone and has a significant impact on health care [1]. The pathogenesis of OA involves whole joint tissues, including the articular cartilage, subchondral bone, meniscus, synovial membrane, and infrapatellar fat pad. Many factors can be characterized for OA, such as synovitis, joint space narrowing, degeneration of articular cartilage and meniscus, subchondral bone remodeling, inflammation of the infrapatellar fat pad, and fibrosis [2,3]. The expression of proinflammatory cytokines including interleukin 1β (IL1β) and tumor necrosis factor α (TNFα) is important in the swollen synovium in OA. These key factors are accompanied by the increased expression of cytokine receptors and reduced expression of inhibitory proteins. These cytokines correlate with cartilage damage by the up regulation of inflammatory or catabolic genes as well as the down regulation of anti-inflammatory or anabolic genes in articular cartilage [4,5]. Particularly, the expression of SOX9, type II collagen, matrix metalloproteinases (MMPs), prostaglandin E2 (PGE2), cytokines, chemokines, reactive oxygen species, and nitric oxide (NO) are affected by IL1β [6,7,8]. During OA, inflammation induces the increase in terminally differentiating chondrocytes and produces high expression of MMP13 to degrade the collagen matrix, resulting in apoptosis of hypertrophic chondrocytes [9]. The inflammation induced by BMPs and cytokines of the joint causes a dysregulated expression of catabolic (cartilaginous matrix proteins) and anabolic proteins (MMPs) to destabilized cartilage homeostasis [10,11]. BMP2 and IL1 are reported to promote the expression of MMP13 in OA chondrocytes [11]. The literature has reported that the source of NO and oxidative stress is the chondrocytes of osteoarthritic articular cartilage [12]. Therefore, NO, IL1β, and TNFα play a key role in suppression of glycosaminoglycan and collagen synthesis, expression of MMPs, and activation of proenzymes in the OA knee [13].

Mesenchymal stem cells (MSCs) have been investigated for use in cell-based therapy for early OA in pre-clinical and clinical treatments [14,15]. A pilot study using autologous bone marrow-derived mesenchymal stem cells on knee OA has shown promising results [16]. Further studies have shown that stem cells secrete a wide range of specific factors to induce paracrine effects on other cells or tissues. Therefore, researchers have used conditioned medium from stem cells, which has potential therapeutic applications in tissue repair in different organs [17,18]. Adipose-tissue-derived mesenchymal stem cells (ADSCs) can be easily expanded and extensively cultivated for use in tissue regeneration or immunomodulation [19,20]. ADSCs can be isolated not only from the abdominal adipose tissue but also from the infrapatellar fat pad [21,22]. ADSCs respond to the inflammatory environment through paracrine factors and modulate the expression of immunosuppressive factors in target cells. There are many paracrine factors involved in the production of signaling in cells, including TGF-β1, IL10, IL2, C–C motif chemokine ligands (CCLs), NO, fibroblast growth factor (FGF), hepatocyte growth factor (HGF), insulin-like growth factor 1 (IGF1), vascular endothelial growth factor (VEGF), platelet-derived growth factor (PDGF), microvesicles, and exosomes [18,23].

Several studies have investigated the mechanisms of conditioned medium (CM) in OA treatment, and investigators have isolated and characterized ADSCs from human adipose tissue [17,24,25]. CM was then collected and used to isolate the microvesicles and exosomes. These extracellular vesicles and CM were used to treat chondrocytes from patients with OA. The results showed that CM, microvesicles, and exosomes reduced IL1β-induced inflammatory mediator production, including TNF-α, IL6, PGE2, and NO [24,26]. Additionally, OA chondrocytes showed significantly decreased activity of MMP and MMP13 expression and increased expression of IL10 and collagen type II after treatment with CM or extracellular vesicles [24]. The effects of CM alone are similar to those of microvesicles and exosomes [24]. Moreover, CM can decrease IL1β-induced inflammatory effects in cartilage and synovium co-culture systems, and reduce the expression levels of NO, MMP13, and a disintegrin and metalloproteinase with thrombospondin motifs 5 (ADAMTS5) [27]. A recent study showed that allogeneic canine CM was injected into both elbow joints in dogs with OA [28]. CM significantly improved the range of motion, and no severe adverse events were observed. This indicates that CM is a safe, cell-free based therapy and a noninvasive therapeutic tool for pain management in knee OA [28].

In this study, pro-inflammatory cytokines and inflammation-induced BMPs were assessed in the articular cartilage of knee OA after ADSC-CM and shockwave therapy. The comparison of ADSC-CM treatment and noninvasive shockwave therapy showed that ADSC-CM down-regulated pro-inflammatory cytokines and inflammatory factors, and up-regulated the expression of the pivotal cartilage-specific extracellular matrix transcription factor SOX9 and type II collagen expression in OA articular cartilage. ADSC-CM therapy may offer several advantages for future clinical translation.

## 2. Materials and Methods

### 2.1. Animals for Experiments

Forty Sprague-Dawley rats (8 weeks old, BioLasco, New Taipei City, Taiwan) were purchased and used for the experiments in this study. The IACUC protocol of the animal study was approved by the Animal Care Committee of Kaohsiung Chang Gung Memorial Hospital (IACUC: 2018031602) on March 16, 2019. The animals were cared for at the Center for Laboratory Animals in the hospital for 1 week before the experiments. All rats were housed at 23 ± 1 °C, humidity at 50 ± 20%, and under light on at 5 am and light off at 5 pm, with a 12-h light and dark cycle as well as supplied food and water.

### 2.2. Animal Model for Knee OA

The rats were anesthetized using Zoletil (25 mg/kg) (tiletamine–zolazepam, Virbac, France) and Rompun (5–10 mg/kg) (xylazine-hydrochloride, Bayer, Leverkusen, Germany), and the left knee of the rat was prepared in a surgically sterile fashion. Through medial parapatellar mini-arthrotomy, the anterior cruciate ligament fibers were transected with a scalpel, and medial meniscectomy was performed by excising the entire medial meniscus to create knee OA. The left knee joint was irrigated, and the incision was closed. Prophylactic antibiotic treatment with ampicillin (25 mg/kg) and pain reduction with ketorolac (1 mg/kg/day) were administered for 5 days after surgery. After surgery, the animals were returned to the housing cage (two rats per cage) and were cared for by a veterinary physician at the Center for Laboratory Animals. The surgical wound and animal activities were monitored daily.

### 2.3. Study Design

Forty rats were randomized into five groups, as shown in Figure 1. In the sham group, rats did not undergo surgery or receive treatment. Rats in the OA group underwent anterior cruciate ligament transection (ACLT) and medial meniscectomy (MMx) of the left knee to create knee OA. In the SW group, the left knees of OA rats were treated with SW (0.25 mJ/mm^2^ with 800 impulses, 4 Hz) at one-week post-surgery. In the CM1 group, the rats received intra-articular injection of 100 μL ADSC-CM into the OA knee at one and three weeks post-surgery. In the CM2 group, the rats received intra-articular injection of 200 μL ADSC-CM at one- and three-weeks post-surgery.

### 2.4. Shockwave Application

The SW group animals received SW treatment using an SW device DUOLITH SD1 (STORZ MEDICAL AG, Tägerwilen, Switzerland). A shockwave was applied to the subchondral bone of the medial tibial condyle of the left knee at 0.5 cm below the joint line and 0.5 cm from the medial skin surface [29]. Each knee received 800 pulses of SW at 0.25 mJ/mm^2^ energy flux density of 4 Hz. After SW therapy, the animals were returned to their housing cages for routine care and observation.

### 2.5. Isolation of Rat Adipose-Derived MSCs

Preparation of rat adipose-derived MSCs (ADSCs) was performed as described in our previous study [29,30]. Briefly, rat ADSCs were isolated from subcutaneous adipose tissues of SD rats at 8 weeks of age. The adipose tissue was minced into one-gram pieces and digested in 3 mL of 0.1% collagenase type I buffer (GIBCO, Waltham, MA, USA) at 37 °C for 2 h. After digestion, 3 mL of Dulbecco’s modified minimal essential medium (DMEM) containing 10% fetal bovine serum (ThermoFisher Scientific, Cleveland, OH, USA) was added. The cell mixture was filtered through a 100 μm filter (BD Biosciences, San Jose, CA, USA) to remove aggregates. The filtrate was centrifuged at 2000 rpm for 5 min at 25 °C, and the pellet was suspended in 1 mL of lysis buffer (Promega, Waldorf, Germany) for 10 min. The mixture was washed with 10 mL of 1% antibiotic-antimycotic mixture in PBS and centrifuged at 2000 rpm for 5 min. Finally, the cell pellet was suspended in complete medium (DMEM with 20% fetal bovine serum, 1% antibiotic antimycotic solution) in a 25 cm^2^ culture flask and maintained in an incubator supplied with a humidified atmosphere of 5% CO_2_ at 37 °C. The rat ADSCs were passaged three times. The third passage of ADSC was observed for the cell morphology and identification of the cell markers. Finally, the ADSCs were used for conditioned medium preparation.

### 2.6. ADSC Phenotyping

The method for the identification of rat ADSCs was performed as described in our previous study [29,30]. The spindle-shaped rat ADSCs were cultured after three to five passages, and the specific surface markers were characterized by flow cytometry (BD LSRII, San Jose, CA, USA). Cells were incubated with specific antibodies conjugated with fluorescein isothiocyanate (FITC) or phycoerythrin (PE) against the indicated markers including CD29, CD45, CD90, CD106, RT1a, and RT1b [29].

### 2.7. Production of ADSC-CM and Intra-Articular Injection of ADSC-CM

The third passage of rat ADSCs was cultured in a T75 flask at a density of 2 × 10^4^ cells/cm^2^. After culturing to 80–90% confluence, ADSCs were washed twice with 1 × PBS buffer and then cultured with serum-free DMEM/F12 (GIBCO, Waltham, MA, USA). ADSCs were then incubated for 48 h with 12 mL serum-free DMEM/F12 at 37 °C and 5% CO_2_. The fresh serum-free DMEM/F12 with no cells was used as a control. After 48 h, the medium was collected and centrifuged for 5 min at 1200× *g* to remove cell debris. The supernatant was transferred to an Amicon Ultra 15 filter (3 kDa cut-off membranes) (Millipore, Bedford, MA, USA) and centrifuged at 4000× *g* for 40 min at 4 °C. Filters were flushed repeatedly with the supernatant, and concentrated ADSC-CM was stored at −80 °C. We repeated the procedure as described above and collected about 20 mL concentrated ADSC-CM as one batch and a total of four different batches were collected. Finally, the ADSC-CM was divided equal amounts into 20 tubes for each batch and using for the experiments. The intra-articular injections of ADSC-CM from different batches were preceded in the CM1 and CM2 groups (two rats per batch; eight rats per group). One hundred microliters of ADSC-CM in the CM1 group and 200 μL of ADSC-CM in the CM2 group were intra-articularly injected into the left rat knee by ultrasound guidance [30].

### 2.8. OARSI Score

The degenerative changes in the cartilage were graded histologically using the Osteoarthritis Research Society International (OARSI) cartilage OA grading system via safranin-O staining [30,31]. The scores were obtained on a scale of 0 to 24 by multiplying the index of grades with stages [30].

### 2.9. Synovitis Scoring

The tissue was stained with hematoxylin and eosin to evaluate synovitis score via assessing thickening of the synovial lining, cellular hyperplasia, and infiltration into the joint cavity and synovium. The three features of synovitis were measured for histopathological assessment, and the score ranks were defined as follows: (1) 0 to 1 indicates no synovitis; (2) 2 to 4 is low-grade synovitis; and (3) 5 to 9 is high-grade synovitis [30].

### 2.10. Histopathological Examination

The left rat knees were subjected to histopathological examination. The harvested left knee specimens were fixed in 4% PBS-buffered formaldehyde at 4 °C for 2 days and decalcified in 10% PBS-buffered EDTA at 4 °C for one month. The decalcified per knee of animal specimens were fixed and subjected to paraffin wax embedding and dissection into 5 μm-thick sections. The one specimen section of knee per animal was stained with hematoxylin-eosin, Safranin-O and immunohistochemical stain (total 8 animals for 8 slide sections in one molecular marker). The degenerative changes in the cartilage were graded histologically using OARSI scores for the assessment of cartilage structure, cartilage cells, and tidemark integrity.

### 2.11. Terminal Deoxynucleotidyl Transferase dUTP Nick End Labeling Assay

The specimen sections were analyzed for apoptosis by terminal deoxynucleotidyl transferase dUTP nick end labeling (TUNEL) assay. TUNEL activity was measured using in situ cell death detection kits (Roche Diagnostic, Penzberg, Germany), according to the manufacturer’s instructions, to identify apoptotic cells in the tissue [32]. The specimens were incubated with peroxidase-conjugated anti-digoxigenin antibody (Roche Diagnostics, Penzberg, Germany). Staining was performed, and a peroxidase substrate (Sigma-Aldrich, Saint Louis, MO, USA) was used to present the color of the TUNEL reaction.

### 2.12. Immunohistochemical Analysis

The specimen sections were analyzed by immunohistochemical methods to determine the level of IL1β (Abcam, San Francisco, CA, USA, Ab-9787, 1:200), MMP13 (Abcam, ab75606, 1:100), type II collagen (Santa Cruz Biotechnology, CA, USA, Sc-52658, 1:100), SOX9 (Abcam, ab26416, 1:100), BMP2 (Abcam, ab6285, 1:100), BMP5 (ThermoFisher Scientific, Cleveland, OH, USA, PA5-97037, 1:100), and BMP6 (ThermoFisher Scientific, Cleveland, OH, USA, PA5-75427, 1:100). The harvested specimens were fixed in 4% PBS-buffered formaldehyde for 2 days and decalcified in PBS-buffered 10% EDTA solution for one month. Decalcified paraffin-embedded samples were cut into 5-μm-thick sections and transferred onto polylysine-coated slides. Sections of the specimens were immunostained with antibodies to identify the protein markers. The immunoreactivity of the specimens was demonstrated using a horseradish peroxidase (HRP)-3′-,3′-diaminobenzidine (DAB) cell and tissue staining kit (R&D Systems, Minneapolis, MN, USA). The immunolabeled positive cells were quantified from five areas in three sections of the same specimen using a Zeiss Axioskop 2 plus microscope (Carl Zeiss, Berlin, Germany). All images were captured using a Cool CCD camera (SNAP-Pro c.f. Digital kit; Media Cybernetics, Carlsbad, CA, USA), and data were analyzed using Image-Pro^®^ Plus image analysis software (Media Cybernetics, Carlsbad, CA, USA).

### 2.13. Statistical Analysis

SPSS (version 26.0, SPSS Inc., Chicago, IL, USA) was used for statistical analysis. Kruskal–Wallis test is a non-parametric method for analyzing two or more independent samples. In this study, using Dunn-Bonferroni nonparametric comparison for post hoc which the Kruskal–Wallis test was significant. All statistical significance was set at *p* < 0.05, 0.01, and 0.001.

## 3. Results

### 3.1. ADSC-CM Protected the Extracellular Matrix and Chondrocytes of the Articular Cartilage in Rat Knee OA

The experimental design is shown in Figure 1. The ADSC-CM and SW therapy groups were compared with the sham and OA groups (Figure 2). Pathological changes in the articular cartilage of the tibia in rat knee OA were observed, including loss of extracellular matrix and the formation of fibrosis in the hyaline cartilage (Figure 2A, OA group). After ADSC-CM and SW treatment, the damage to hyaline articular tissue was obviously improved, as shown by safranin-O staining (Figure 2A, SW, CM1, and CM2 groups). For the OARSI cartilage scores, the CM2 and SW groups showed significant improvement in the repair of hyaline cartilage as compared with the OA group (Figure 2B; *p* < 0.05). In addition, among the treatment groups, the improvement in the articular cartilage in the SW and CM2 groups was better than that of the CM1 group (*p* < 0.01). Furthermore, the results showed that CM2 improved the loss of extracellular matrix and articular chondrocytes than CM1 in trends but no significance.

### 3.2. The Improvement of Synovitis in Knee OA

Synovitis of the rat joint was measured after the treatment (Figure 3). The enlargement of the lining cell layer and infiltration of mononuclear cells and neutrophils were improved in the synovial membrane of the knee after treatment (Figure 3A, white arrow). The SW (4.33 ± 0.422, *p* < 0.001), CM1 (5.66 ± 0.422, *p* < 0.05), and CM2 (4.00 ± 0.931, *p* < 0.05) groups showed significant improvement in synovitis compared with the OA group (7.33 ± 0.558) (Figure 3B). Among the treatment groups, the CM2 and SW groups showed better improvement in trends than the CM1 group but no significance.

### 3.3. ADSC-CM Down-Regulates the Inflammatory Molecular Factors in the OA Knee

Inflammatory cytokines that induce MMPs are important for living chondrocytes and the composition of articular cartilage of the knee. In the study, the inflammatory cytokine IL1β, apoptotic cells, and MMP13 in OA rat knees were assessed after treatment (Figure 4). The expression of IL1β was significantly reduced in the SW (6.74 ± 0.532, *p* < 0.05), CM1 (4.77 ± 0.498, *p* < 0.001), and CM2 (3.76 ± 0.264, *p* < 0.001) groups compared with that in the OA group (10.823 ± 0.583) (Figure 4A). In addition, the CM2 (3.76 ± 0.264, *p* < 0.05) group showed the greatest reduction in the expression of ILβ compared to SW (6.74 ± 0.532) and CM1 (4.77 ± 0.498, *p* < 0.05) groups. The cell death marker, TUNEL activity, was measured (Figure 4B). The TUNEL activity was reduced in the SW (3.98 ± 0.341, *p* < 0.01), and CM2 (4.07 ± 0.245, *p* < 0.01) groups compared with that in the OA (7.38 ± 0.46) group. Furthermore, the SW (3.07 ± 0.541, *p* < 0.05), CM1 (2.11 ± 0.094, *p* < 0.001), and CM2 (1.77 ± 0.221, *p* < 0.001) groups showed significantly reduced expression of MMP13 as compared with the OA (5.57 ± 0.392) group (Figure 4C).

### 3.4. ADSC-CM Up-Regulated Key Molecular Factors for Chondrogenesis and Extracellular Matrix in the Articular Cartilage of OA Knees

The expression of SOX9 and type II collagen was measured after treatment (Figure 5). The expression level of SOX9 was significantly increased in the SW (8.12 ± 0.409, *p* < 0.05), and CM2 (8.73 ± 0.600, *p* < 0.01) groups compared with that of the OA (5.50 ± 0.500) group. In addition, the expression of type II collagen was increased in the treatment groups and CM2 (32.61 ± 1.992, *p* < 0.001) was significantly increased by compared with the OA group (14.06 ± 1.533) (Figure 5B).

### 3.5. ADSC-CM Modulated the Expression of BMP2, BMP5, and BMP6 in the OA Knee Articular Cartilage

Inflammation-induced BMPs are harmful to the articular cartilage during OA development. In the current study, the expression of BMP2, BMP5, and BMP6 was measured in the knee OA with and without treatment (Figure 6). The expression level of BMP2 was significantly higher in the OA group than in the sham group (Figure 6A). After treatment, in the SW (8.51 ± 0.424, *p* < 0.05), CM1 (8.57 ± 0.303, *p* < 0.05), and CM2 (6.15 ± 0.546, *p* < 0.01) groups, BMP2 was significantly reduced as compared with that in the OA (12.91 ± 0.967) group, and the CM2 group showed the greatest reduction in the expression of BMP2. In addition, BMP5 was induced in the OA (7.67 ± 0.399) group and reduced after treatment in the SW (3.66 ± 0.359, *p* < 0.001), CM1 (3.61 ± 0.510, *p* < 0.001), and CM2 (2.41 ± 0.357, *p* < 0.001) groups. Finally, BMP6 was significantly reduced in the SW (2.88 ± 0.104, *p* < 0.001), CM1 (3.86 ± 0.302, *p* < 0.05), and CM2 (2.96 ± 0.242, *p* < 0.001) groups compared with that in the OA (5.03 ± 0.227) group.

## 4. Discussion

In this study, ADSC-CM modulated the inflammatory factors and inflammation-induced BMPs, such as BMP2, BMP5, and BMP6, to prevent damage to the articular cartilage in rat knee OA. In addition, ADSC-CM had a dose-dependent effect and was compared with noninvasive SW therapy in the treatment of knee OA. The results demonstrated that the expression of inflammatory cytokines (IL1-β), apoptosis activity (TUNEL), and markers for OA (MMP13) were induced in knee OA, and were reduced after ADSC-CM and SW therapy. The expression of cartilaginous-specific markers, SOX9 and type II collagen was significantly improved in the articular cartilage of OA rat knees following ADSC-CM and SW treatments.

ADSCs are a type of MSC that are prepared and cultured from adipose tissues and can differentiate into different cell lineages [33]. ADSCs have typical mesenchymal markers including CD13, CD29, CD44, CD63, CD73, CD90, and CD105, and are negative for hematopoietic antigens, CD14, CD31, CD45, and CD144 [34,35,36]. ADSCs can self-renew and differentiate multidirectionally and secrete biological factors, including cytokines, growth factors, exosomes, RNA, and DNA [33,37]. ADSCs have been widely used in clinical applications, tissue engineering, and regenerative medicine [38,39]. However, researchers have revealed that ADSCs cannot survive for a long time when implanted into the human body [40,41]. In addition, the therapeutic effects of ADSCs may be due to their secretome derivatives [27,40]. Therefore, ADSC-CM is considered advantageous over cells because it is easy to prepare, does not require expanding passages, does not undergo spontaneous transformation, is easily handled and stored, and is a quickly available product.

Here, we verified rat ADSCs and cultured them to retain the CM for the experiments. The ADSC-CM and SW, which is a noninvasive acoustic wave that promotes tissue regeneration, were compared for the treatment of rat knee OA. ADSC-CM and SW therapy have been reported to have chondroprotective effects to reduce the expression of IL1β-induced MMP13 and increase the expression of type II collagen in OA chondrocytes in animal models [42,43,44]. Our results showed that ADSC-CM and SW therapy reduced damage to the articular cartilage and synovitis in rat knee OA (Figure 2 and Figure 3). In addition, ADSC-CM was better at reducing the expression of IL1β and MMP13 than SW therapy (Figure 4). The results of this study suggest that ADSC-CM is an alternative efficient method for articular cartilage protection [45]. Synovitis is a condition of inflammation of the synovial membrane. Swelling, stiffness, and pain occur in the knee and further induce knee osteoarthritis [46]. When the synovial membrane is inflamed, neutrophils, fibroblast-like synoviocytes, and macrophage-like synoviocytes are activated and infiltrate into the inflamed membrane [47]. Studies have reported that ADSC-CM regulates the expression of pro-inflammatory cytokines CXCL1, IL1β, and IL6 as well as the anti-inflammatory cytokine IL10 in the treatment of inflammatory diseases [42,48]. Therefore, ADSC-CM has anti-inflammatory effects and modulates inflammation in the synovial membrane of OA through regulation of the immune response in immune cells such as macrophages and neutrophils [25,42,49].

BMPs are reported to play an important role in the repair of articular cartilage; however, the harmful effects of over-induction of BMPs have also been reported in inflammatory joint disease [9,50]. The expression profiles of BMP2, BMP5, and BMP6 were observed to increase in knee OA and decrease after ADSC-CM and SW treatments (Figure 6). In addition, the expression of BMP2, BMP6, and BMP7 is increased in the osteoarthritic cartilage of humans and animals, and cultured OA chondrocytes [50,51,52]. In our study, the expression of IL1, MMP13, BMP2, BMP5, and BMP6 in the rat OA knee was higher than that of the sham control and reduced to sham levels after ADSC-CM and SW treatments (Figure 4 and Figure 6). Furthermore, ADSC-CM and SW therapy enhanced the expression of SOX9 and type II collagen in the knee OA (Figure 5). Endogenous BMPs can stimulate proteoglycan synthesis and are involved in the homeostasis and maintenance of joint integrity during injury. However, severe inflammation (for example, excessive IL1) induced high expression of endogenous BMPs, which might disrupt the balance in cartilage homeostasis [4,9,52]. The study showed that ADSC-CM was more effective than SW in modulating the expression of catabolic and anabolic factors in the rat knee OA (Figure 4, Figure 5, and Figure 6). However, the detailed mechanism by which endogenous BMPs modulate cartilage homeostasis in OA is still unclear. Further studies may display clearly defined functions and signaling pathways of these fascinating growth factors in OA progression, repair, and treatment.

The limitations of this study are as follows: First, the study was a small animal experiment, and the results would be different from those of large animal or human clinical trials. Second, the pathological changes in rat knee OA might be different from those in human knee OA and require further investigation. Third, there are different methods for the preparation of ADSC-CM, which may result in different contents of growth factors, secretomes, extracellular vesicles, DNA, and RNA. The products for GMP grade ADSC-CM should be considered and further established. Fourth, the brands of the shockwave devices on the market and their effects on knee OA might also be different. Fifth, the dosage of ADSC-CM and shockwave were based on previous animal studies and may not be the optimal doses for human clinical treatment. Sixth, ADSC-CM from rats is different from that of humans.

## 5. Conclusions

In the study, rat ADSC-CM and SW therapy can protect against the loss of extracellular matrix in the articular cartilage and improve synovitis in knee OA. In addition, rat ADSC-CM and SW therapy reduced the expression of IL1β, TUNEL activity, and MMP13, and promoted the expression of SOX9 and type II collagen in the articular cartilage of the knee OA. The harmful, inflammation-induced BMP2, BMP5, and BMP6 are modulated after treatments. Furthermore, a two-fold dosage of rat ADSC-CM was better than one dosage of rat ADSC-CM and SW therapy in the treatment of knee OA. Taken together, this study provides significant findings regarding ADSC-CM in comparison with SW therapy for the treatment of knee OA for future clinical trials.

## Figures and Tables

**Figure 1 biomedicines-09-01399-f001:**
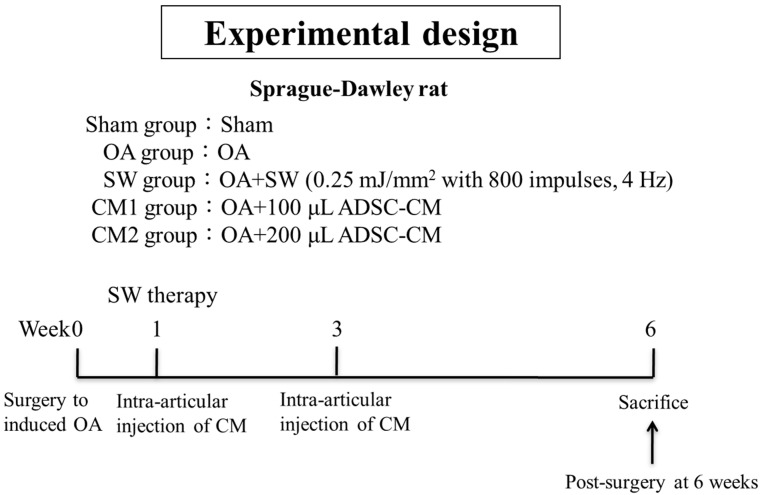
The study design. The graph displays the study design of the experiment including knee surgery (osteoarthritis; OA), shockwave (SW) application, 100 and 200 μL adipose-derived mesenchymal stem cells-conditioned medium (ADSC-CM) injections at 1- and 3-weeks post-surgery, respectively, and sacrifice of animals at 6 weeks post-surgery. Eight rats were used for each group.

**Figure 2 biomedicines-09-01399-f002:**
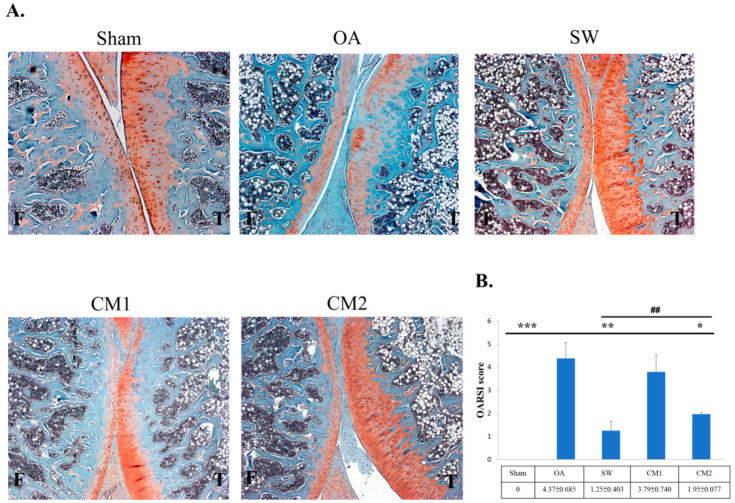
The pathological changes of the knee after shockwave (SW) and adipose-derived mesenchymal stem cells-conditioned medium (ADSC-CM) treatment. (**A**) The articular cartilage degradation of the knee in different groups (50× magnification). The articular cartilage damage is evident in the OA group, and improvement was observed in SW, CM1, and CM2 groups. (**B**) The OARSI score in each group. * *p* < 0.05, ** *p* < 0.01 and *** *p* < 0.001 as comparing sham, SW, CM1, and CM2 groups with the OA group. ^##^
*p* < 0.01 as comparing CM1 and CM2 groups with the SW group. The 100 and 200 μL adipose-derived mesenchymal stem cells-conditioned medium treatments are indicated as CM1 and CM2, respectively. Eight rats were used for each group.

**Figure 3 biomedicines-09-01399-f003:**
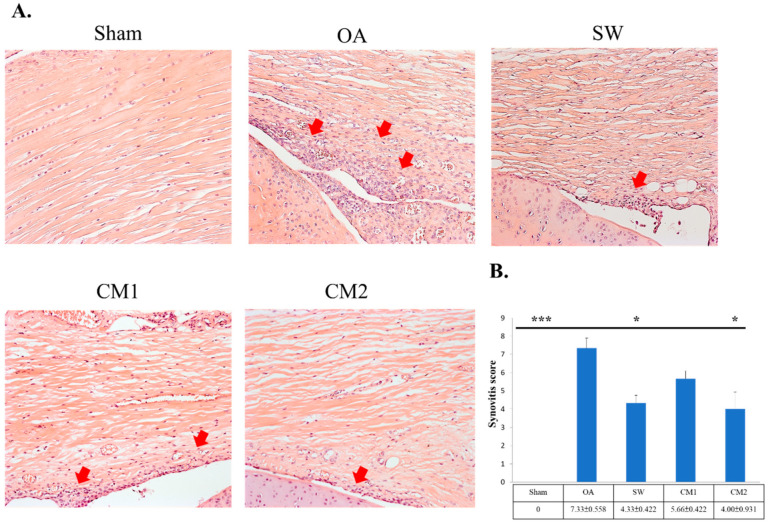
Evaluation of histological changes in the synovium membrane of rats with osteoarthritis (OA) after treatments. (**A**) The sections of synovium were observed by hematoxylin and eosin staining with 200× magnification. The lining cell layer, infiltration of mononuclear cells, and neutrophils are indicated by arrows. (**B**) The synovitis scores were calculated for each group. * *p* < 0.05 and *** *p* < 0.001 as compared with the OA group. Eight rats were used for each group. Osteoarthritis is indicated as OA. Shockwave is indicated as SW. The 100 and 200 μL adipose-derived mesenchymal stem cells-conditioned medium groups are indicated as CM1 and CM2, respectively.

**Figure 4 biomedicines-09-01399-f004:**
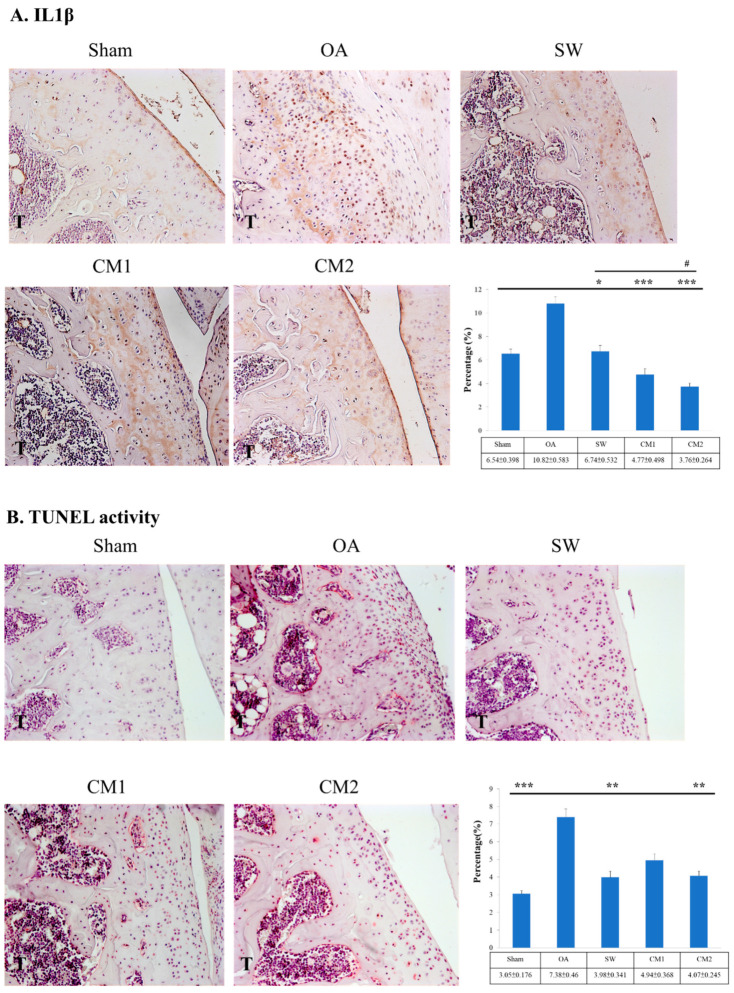
Apoptosis and physiological turnover of cartilage extracellular matrix factors in OA treatments. (**A**) Immunohistochemical analysis for IL1β in the experiments (×400 magnification). The data show the percentage of IL1β expression in the articular cartilage of the tibia for each group in the left panel. (**B**) The activity of TUNEL (×400 magnification). (**C**) The expression level of MMP13 (×400 magnification). * *p* < 0.05, ** *p* < 0.01 and *** *p* < 0.001 as compared with the OA group. ^#^
*p* < 0.05 as compared with the SW group. Eight rats were used for each group.

**Figure 5 biomedicines-09-01399-f005:**
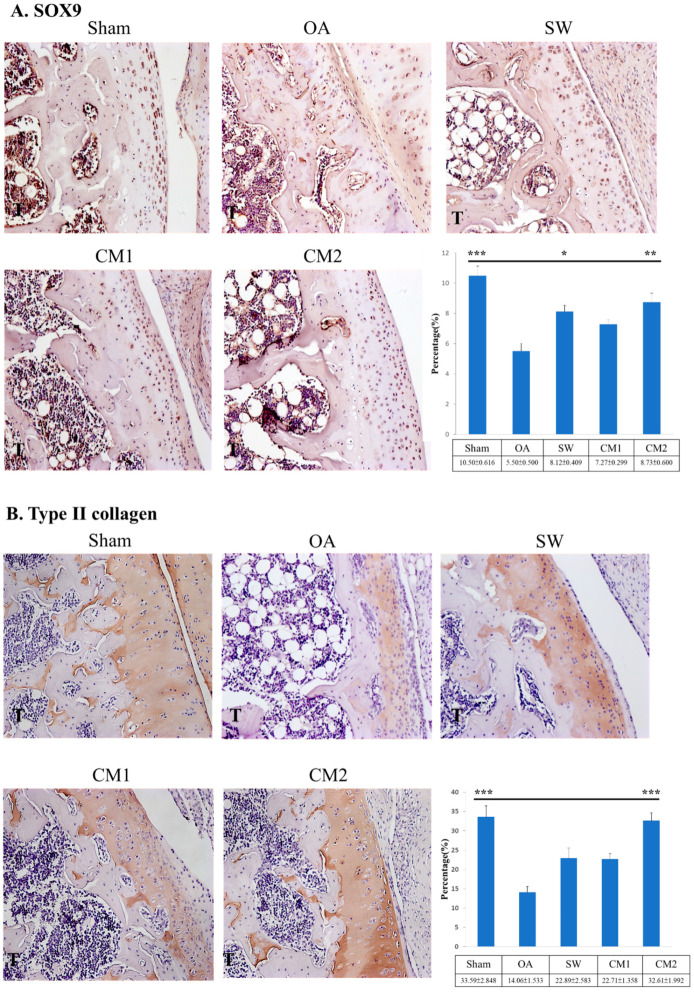
Immunohistochemical analysis for (**A**) SOX9 and (**B**) type II collagen (×400 magnification), and the percentage of SOX9 and type II collagen expression in the articular cartilage of the tibia for each group. * *p* < 0.05, ** *p* < 0.01 and *** *p* < 0.001 as compared with the osteoarthritis (OA) group. Eight rats were used for each group. Shockwave is indicated as SW. Osteoarthritis is indicated as OA. The 100 and 200 μL adipose-derived mesenchymal stem cells-conditioned medium groups are indicated as CM1 and CM2, respectively.

**Figure 6 biomedicines-09-01399-f006:**
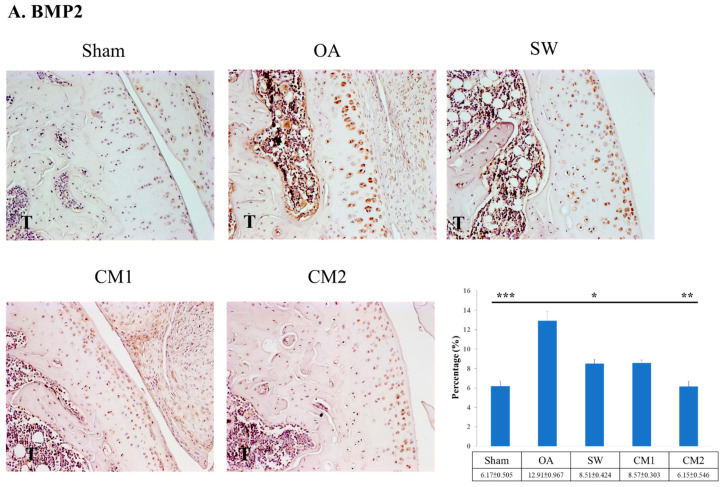
Immunohistochemical analysis for (**A**) BMP2, (**B**) BMP5, and (**C**) BMP6 (×400 magnification), and the percentage of BMP2, BMP5, and BMP6 expression in the articular cartilage of the tibia for each group is shown on the left. The 100 and 200 μL adipose-derived mesenchymal stem cells-conditioned medium groups are indicated as CM1 and CM2, respectively. SW indicates shockwave. * *p* < 0.05, ** *p* < 0.01, and *** *p* < 0.001 as compared with the osteoarthritis (OA) group. Eight rats were used for each group.

## Data Availability

The datasets of present study can be available from the corresponding author upon request.

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
