# Peer review of "Adipose-Derived Mesenchymal Stem Cells-Conditioned Medium Modulates the Expression of Inflammation Induced Bone Morphogenetic Protein-2, -5 and -6 as Well as Compared with Shockwave Therapy on Rat Knee Osteoarthritis"

_biomedicines, 2021, doi:10.3390/biomedicines9101399_

Round 1

Reviewer 1 Report

Biomedicines-1368122.v1    Review Report

English must be revised        Extensive editing of English language and style required

Introduction

The introduction needs to be revised to better explain OA disease and contextualize the contribution of all tissues and related molecules. Furthermore, references have to be updated with the most relevant and recent publications.

  • OA is now considered a disorder involving the whole joint tissues (i.e. cartilage, subchondral bone, meniscus, synovial membrane and infrapatellar fat pad). OA is characterized by cartilage and meniscus degeneration, subchondral bone remodeling, synovial membrane inflammation and infrapatellar fat pad inflammation and fibrosis (doi:1002/art.34453; doi: 10.3390/ijms21176016).
  • Better describe the involvement of the molecules evaluated in the study in the pathogenesis of OA limiting to refer to studies regarding other factors.

  • Many therapies have been developed to treat OA. Cell therapy is used as a regenerative approach with MSCs isolated from different sources including adipose tissue. ADSCs can be isolated not only from the abdominal adipose tissue but also from the infrapatellar fat pad. Please update the references (doi: 1007/s12178-020-09624-0, doi: 10.3389/fcell.2019.00323)

Lines 94-95: better describe the aim of the study.

Lines 96-99: move this part to discussion and conclusion sections.

Materials and methods

Line 121: Figure 1 should be inserted close to its first citation.

Line 130: Please correct. SW prevent and does not induce OA.

Line 167: complete this paragraph explaining how the intra-articular injection of ADSC-CM has been done. 

                Correct the title of the paragraph in accordance to this modification.

Line 169: correct “generative” in degenerative changes.

Line 171: use the original reference for OARSI score (doi: 10.1016/j.joca.2005.07.014)

Lines 190-191 Better describe this assay also explaining the abbreviation.

Results

Only the results must be presented in this section. Any other information should be placed in the discussion.

Line 221: Figure 2 should be inserted close to its first citation. Furthermore, summarize the caption. Place abbreviations at the end of the caption.

For example: “Figure 2. The pathological changes of the knee cartilage after shockwave (SW) and adipose-derived mesenchymal stem cells conditioned medium (ADSC-CM) treatment (n=8). (A) The microphotographs of the knee show cartilage degradation of the knee in the different groups (magnification 50X). The microphotographs of articular cartilage illustrate the changes of cartilage damage is evident in OA group but not and protect in SW, CM1 and CM2 groups. (B) The OARSI score is displayed in the each group. The images are represented at 50× magnification and n=8. §P < 0.05 as comparing CM1 and CM2 treatment groups; *P < 0.05 and ***P < 0.01 comparing SW, CM1 and CM2 groups with OA group.”

# add explanation for this symbol

Line 221: “The ADSC-CM was injected into the rat knee OA by ultrasound guidance” – this info should be placed in materials and methods section.

Lines 223 and 227-230: show only the results explaining if they are present/better or not comparing the different groups. Other info should be placed in the discussion.

Line 232: Figure 3 should be inserted close to its first citation. Its caption should be placed under the figure.

Line 244: Figure 4 is lacking and should be inserted close to its first citation

Line 259: Figure 5 should be inserted close to its first citation.

Line 259-260 and 263: This data should be placed in materials and methods section.

Line 273: Figure 6 should be inserted close to its first citation.

Lines 285-286: This part should be deleted.

Discussion

In the discussion there are some parts that should be placed in the introduction. Please report only the studies and the information useful to explain your results.

Line 345: Ref 32. Use references of clinical applications in OA and not in multiple sclerosis.

Author Response

Review 1:

Biomedicines-1368122.v1    Review Report

English must be revised  Extensive editing of English language and style required

Response: The manuscript is English editing by Editage.

 Introduction

The introduction needs to be revised to better explain OA disease and contextualize the contribution of all tissues and related molecules. Furthermore, references have to be updated with the most relevant and recent publications.

  • OA is now considered a disorder involving the whole joint tissues (i.e. cartilage, subchondral bone, meniscus, synovial membrane and infrapatellar fat pad). OA is characterized by cartilage and meniscus degeneration, subchondral bone remodeling, synovial membrane inflammation and infrapatellar fat pad inflammation and fibrosis (doi:1002/art.34453; doi: 10.3390/ijms21176016).

Response: Thanks reviewer’s suggestion. We verified the text and cited the reference in the Introduction and showed as following: The pathogenesis of OA involves whole joint tissues, including the articular cartilage, subchondral bone, meniscus, synovial membrane, and infrapatellar fat pad. Many factors can be characterized for OA, such as synovitis, joint space narrowing, degeneration of articular cartilage and meniscus, subchondral bone remodeling, inflammation of the infrapatellar fat pad, and fibrosis [2,3].

  • Better describe the involvement of the molecules evaluated in the study in the pathogenesis of OA limiting to refer to studies regarding other factors.

Response: Thank you very much. We described the involvement of the molecules evaluated in the study to the section of introduction.

  • Many therapies have been developed to treat OA. Cell therapy is used as a regenerative approach with MSCs isolated from different sources including adipose tissue. ADSCs can be isolated not only from the abdominal adipose tissue but also from the infrapatellar fat pad. Please update the references (doi: 1007/s12178-020-09624-0, doi: 10.3389/fcell.2019.00323)

Response: Thanks the comments. We verified the text and cited the reference in the Introduction as following: ADSCs can be isolated not only from the abdominal adipose tissue but also from the infrapatellar fat pad [22,23].

Lines 94-95: better describe the aim of the study and Lines 96-99 move this part to discussion and conclusion sections.

Response: Thanks the comments. We verified the text in the Introduction as the following: In this study, pro-inflammatory cytokines and inflammation-induced BMPs were assessed in the articular cartilage of knee OA after ADSC-CM and shockwave therapy. The comparison of ADSC-CM treatment and noninvasive shockwave therapy showed that ADSC-CM down-regulated pro-inflammatory cytokines and inflammatory factors, and up-regulated the expression of the pivotal cartilage-specific extracellular matrix transcription factor SOX9 and type II collagen expression in OA articular cartilage. ADSC-CM therapy may offer several advantages for future clinical translation.

Materials and methods

Line 121: Figure 1 should be inserted close to its first citation.

Response: Thanks reviewer’s suggestion. We used the template of the journal and put the figure1 in the Section 3.2. Figures, Tables and Schemes.

Line 130: Please correct. SW prevent and does not induce OA.

Response: Thanks reviewer’s suggestion. We revised the sentence as following: The SW group animals received SW treatment using an SW device DUOLITH SD1 (STORZ MEDICAL AG, Tägerwilen, Switzerland).  

Line 167: complete this paragraph explaining how the intra-articular injection of ADSC-CM has been done. Correct the title of the paragraph in accordance to this modification.

Response: Thank you very much for the reviewer’s suggestion. We verified the paragraph and correct the title as the following:

2.7. Production of ADSC-CM and intra-articular injection of ADSC-CM

The rat ADSCs were cultured in T75 flasks at a density of 2 × 104 cells/cm2. After culturing to 80–90% confluence, ADSCs were washed with 1 × PBS buffer and then cultured with serum-free DMEM/F12 (Gibco BRL, USA). Flasks were incubated for 48 h with 12 mL serum-free DMEM/F12 at 37 °C and 5% CO2. Serum-free DMEM/F12 with no cells was used as a control. After 48 h, the medium was removed and centrifuged for 5 min at 1200 × g to remove cell debris. The supernatant was transferred to an Amicon Ultra 15 filter (3 kDa cut-off membranes) (Millipore, USA) and centrifuged at 4000 × g for 40 min at 4 °C. Filters were flushed repeatedly with the supernatant, and concentrated ADSC-CM was stored at -80 °C for subsequent experiments.

The intra-articular injection of ADSC-CM proceeded in the CM1 and CM2 groups. One hundred microliters of ADSC-CM in the CM1 group and 200 μL of ADSC-CM in the CM2 group were intra-articularly injected into the left rat knee by ultrasound guidance [31].

Line 169: correct “generative” in degenerative changes.

Response: We verified the typo in this paragraph.

Line 171: use the original reference for OARSI score (doi: 10.1016/j.joca.2005.07.014)

Response: Thanks suggestion. The original reference was citation in the text.

Lines 190-191 Better describe this assay also explaining the abbreviation.

 Response: We verified and described the terminal deoxynucleotidyl transferase dUTP nick end labeling (TUNEL) assay in the Materials and Methods as bellowing:

2.11. Terminal deoxynucleotidyl transferase dUTP nick end labeling assay

The specimens were analyzed for apoptosis by terminal deoxynucleotidyl transferase dUTP nick end labeling (TUNEL) assay. TUNEL activity was measured using in situ cell death detection kits (Roche Diagnostic, Germany), according to the manufacturer's instructions, to identify apoptotic cells in the tissue [33]. The specimens were incubated with peroxidase-conjugated anti-digoxigenin antibody (Roche Diagnostics, Germany). Staining was performed, and a peroxidase substrate (Sigma-Aldrich, USA) was used to present the color of the TUNEL reaction.

Results

Only the results must be presented in this section. Any other information should be placed in the discussion.

Line 221: Figure 2 should be inserted close to its first citation. Furthermore, summarize the caption. Place abbreviations at the end of the caption.

For example: “Figure 2. The pathological changes of the knee cartilage after shockwave (SW) and adipose-derived mesenchymal stem cells conditioned medium (ADSC-CM) treatment (n=8). (A) The microphotographs of the knee show cartilage degradation of the knee in the different groups (magnification 50X). The microphotographs of articular cartilage illustrate the changes of cartilage damage is evident in OA group but not and protect in SW, CM1 and CM2 groups. (B) The OARSI score is displayed in the each group. The images are represented at 50× magnification and n=8. §P < 0.05 as comparing CM1 and CM2 treatment groups; *P < 0.05 and ***P < 0.01 comparing SW, CM1 and CM2 groups with OA group.”

# add explanation for this symbol

Response: Thanks reviewer’s suggestion. The figures were inserted in the format of the journal. We verified the figure legend as the suggestion of reviewer and place abbreviations at the end of the caption. The verified legend as following:

Figure 2. The pathological changes of the knee after shockwave (SW) and adipose-derived mesenchymal stem cells-conditioned medium (ADSC-CM) treatment (N=8). (A) The articular cartilage degradation of the knee in different groups (50× magnification). The articular cartilage damage is evident in the OA group, and improvement was observed in SW, CM1, and CM2 groups. (B) The OARSI score in each group. *P < 0.05 and ***P < 0.001 comparing sham, SW, CM1, and CM2 groups with the OA group. #P < 0.05 comparing CM1 and CM2 groups with the SW group. §P < 0.05 comparing CM1 with CM2. Osteoarthritis is indicated as OA. The 100 and 200 μL adipose-derived mesenchymal stem cells-conditioned medium treatments are indicated as CM1 and CM2, respectively.

Line 221: “The ADSC-CM was injected into the rat knee OA by ultrasound guidance” – this info should be placed in materials and methods section.

Response: Thanks reviewer’s suggestion. We verified the information into the Section 2.7. Production of ADSC-CM and intra-articular injection of ADSC-CM.

Lines 223 and 227-230: show only the results explaining if they are present/better or not comparing the different groups. Other info should be placed in the discussion.

Response: Thanks reviewer’s suggestion. We verified the paragraph as bellowing:

3.1. ADSC-CM protected the extracellular matrix and chondrocytes of the articular cartilage in rat knee OA

The experimental design is shown in Fig. 1. The ADSC-CM and SW therapy groups were compared with the sham and OA groups (Figure 2). Pathological changes in the articular cartilage of the tibia in rat knee OA were observed, including loss of extracellular matrix and the formation of fibrosis in the hyaline cartilage (Figure 2A, OA group). After ADSC-CM and SW treatment, the damage to hyaline articular tissue was obviously improved, as shown by safranin-O staining (Figure 2A, SW, CM1, and CM2 groups). For the OARSI cartilage scores, the CM2 and SW groups showed signifi-cant improvement in the repair of hyaline cartilage as compared with the OA group (Figure 2B; P < 0.05). In addition, among the treatment groups, the improvement in the articular cartilage in the SW and CM2 groups was better than that of the CM1 group (P < 0.05). Furthermore, the results showed that CM2 improved the loss of extracellular matrix and articular chondrocytes more than CM1 (P < 0.05), and a dose-dependent ef-fect of ADSC-CM was demonstrated. 

Line 232: Figure 3 should be inserted close to its first citation. Its caption should be placed under the figure.

Response: Thanks reviewer’s suggestion. The figures were inserted in the format of the journal. The caption of figure 3 was verified.

Line 244: Figure 4 is lacking and should be inserted close to its first citation

Response: Thanks reviewer’s suggestion. All figures were inserted close to its first citation.

Line 259: Figure 5 should be inserted close to its first citation.

Response: Thanks suggestion. All figures were inserted close to its first citation

Line 259-260 and 263: This data should be placed in materials and methods section.

Response: Thanks reviewer’s suggestion. We verified the sentence.

Line 273: Figure 6 should be inserted close to its first citation.

Response: Thanks reviewer’s suggestion. All figures were inserted close to its first citation.

Lines 285-286: This part should be deleted.

Response: We deleted the description.

Discussion

In the discussion there are some parts that should be placed in the introduction. Please report only the studies and the information useful to explain your results.

Line 345: Ref 32. Use references of clinical applications in OA and not in multiple sclerosis.

Response: We verified the discussion and reference as well as cited clinical experiments in OA.

Reviewer 2 Report

The study analyzed the effect of ADSC-CM and shockwave therapy in a rat model of ACLT with MMx knee OA. The authors reported that higher ADSC-CM and SW treatment groups had significant lower OARSI score and synovitis score. Moreover higher ADSC-CM treatment was associated with lower inflammatory markers (IL1beta and TUNEL), MMP13 and BMP2, BMP5 and BMD6. On the other hand, higher ADSC-CM treatment was associated with higher cartilage repairing markers (type II collagene and SOX9). Although interesting I have the following comments for the authors:

  • Page 2, line 33. Please correct “OARSI core”.
  • Page 2, line 33. “increased OARSI (s)core”, was it “decreased”?
  • Page 2, line 35. Please correct “terminal deoxynucleotidyl transferase dUTP nick end labeling and”.
  • I suggest the authors to add a short conclusion to the Abstract.
  • Page 3, line 78. Please define “CM”.
  • Page 5, line 188. Please define “TUNEL”.
  • Page 6, line 211. The authors stated that ANOVA test was used. Were the data in parametric distribution? What was the test used to assess the normal distribution of the five groups of eight rats? If the data were not in normal distribution please use a non parametric test with an appropriate post-test for pairwise comparisons.
  • Page 6, line 212. Why have three p values been defined?
  • Page 6, lines 214-216. Please erase these lines.
  • Page 7, line 286. Please erase this line.
  • Page 14, line 403. Please define “ESWT”.
  • I suggest editing help from someone with full professional proficiency in English.

Author Response

Reviewer2

The study analyzed the effect of ADSC-CM and shockwave therapy in a rat model of ACLT with MMx knee OA. The authors reported that higher ADSC-CM and SW treatment groups had significant lower OARSI score and synovitis score. Moreover higher ADSC-CM treatment was associated with lower inflammatory markers (IL1beta and TUNEL), MMP13 and BMP2, BMP5 and BMD6. On the other hand, higher ADSC-CM treatment was associated with higher cartilage repairing markers (type II collagene and SOX9). Although interesting I have the following comments for the authors:

  • Page 2, line 33. Please correct “OARSI core”.

Response: Thanks reviewer’s suggestion. We corrected the typo.

  • Page 2, line 33. “increased OARSI (s)core”, was it “decreased”?

Response: We verified the sentence as following: The OARSI and synovitis scores of CM2 and SW group were significantly decreased compared with those of the OA group (P<0.05).

  • Page 2, line 35. Please correct “terminal deoxynucleotidyl transferase dUTP nick end labeling and”.

Response: We corrected the typo.

  • I suggest the authors to add a short conclusion to the Abstract.

Response: Thanks reviewer’s suggestion. In conclusion, ADSC-CM and SW ameliorated the expression of inflammatory cytokines and inflammation-induced BMPs to protect the articular cartilage of OA joint.

  • Page 3, line 78. Please define “CM”.

Response: Thanks reviewer’s suggestion. We define the conditioned medium (CM) in the text.

  • Page 5, line 188. Please define “TUNEL”.

Response: Thanks reviewer’s suggestion. We define the terminal deoxynucleotidyl transferase dUTP nick end labeling (TUNEL) assay in the text.

  • Page 6, line 211. The authors stated that ANOVA test was used. Were the data in parametric distribution? What was the test used to assess the normal distribution of the five groups of eight rats? If the data were not in normal distribution please use a non parametric test with an appropriate post-test for pairwise comparisons.Page 6, line 212. Why have three p values been defined?

Response: Thanks the comments. We verified the text as following: SPSS (version 17.0, SPSS Inc., Chicago, IL, USA) was used for statistical analysis. The data of the treatment and sham groups were compared statistically using the Mann-Whitney U test. The data of the different groups were calculated statistically using the Chi square test. All statistical significance was set at P < 0.05, 0.01, and 0.001.

  • Page 6, lines 214-216. Please erase these lines.

Response: We deleted the description.

  • Page 7, line 286. Please erase this line.

Response: We deleted the sentence.

  • Page 14, line 403. Please define “ESWT”.

Response: We verified the typo of ESWT to shockwave therapy.

  • I suggest editing help from someone with full professional proficiency in English.

Response: The manuscript is English editing by Editage.
